# Towards a Stable and High-Performance Hindered Phenol/Polymer-Based Damping Material Through Structure Optimization and Damping Mechanism Revelation

**DOI:** 10.3390/polym11050884

**Published:** 2019-05-15

**Authors:** Kangming Xu, Qiaoman Hu, Junhui Wang, Hongdi Zhou, Jinlei Chen

**Affiliations:** 1College of Materials and Chemical Engineering, Chongqing University of Arts and Sciences, Chongqing 402160, China; 20150064@cqwu.edu.cn (K.X.); 18996682270@163.com (J.W.); HD1412159@163.com (H.Z.); chenjinlei513@163.com (J.C.); 2Research Institute for New Materials Technology, Chongqing University of Arts and Sciences, Chongqing 402160, China; 3College of Chemistry, Sichuan University, Chengdu 610065, China

**Keywords:** hindered phenol/polymer-based hybrid damping materials, stability, high damping performance, structure optimization, damping mechanism

## Abstract

Although hindered phenol/polymer-based hybrid damping materials, with excellent damping performance, attract more and more attention, the poor stability of hindered phenol limits the application of such promising materials. To solve this problem, a linear hindered phenol with amorphous state and low polarity was synthesized and related polyurethane-based hybrid materials were prepared in this study. The structure and state of the hindered phenol were confirmed by nuclear magnetic resonance spectrum, Fourier transform infrared spectroscopy (FT-IR) and X-ray diffraction (XRD). The existence of intermolecular hydrogen bonds (HBs) between hindered phenol and polyurethane was confirmed by FT-IR, and the amorphous state of the hybrids was confirmed by XRD. Moreover, by a combination of molecular dynamics simulation and dynamic mechanical analysis, the relationship between the structure optimization of the hindered phenol and the high damping performance of the hybrids was quantitatively revealed. By constructing the synthetic hindered phenol, the intramolecular HBs between hindered phenols were restricted, while the strength and concentration of the intermolecular HBs increased by increasing the amount of hindered phenol. Thus, intermolecular interactions were enhanced, which lead to the compact chain packing of polyurethane, extended chain packing of hindered phenol, and good dispersion of hindered phenol in polyurethane. Therefore, the stability of the hindered phenol and the damping properties of the hybrids were both improved. The experiment results are expected to provide some useful information for the design and fabrication of high-performance polymeric damping materials.

## 1. Introduction

Nowadays, vibration and noise issues are predominant in engineering systems and affect people’s health [1,2,3]. In order to reduce these pollutants, researchers are devoting time to develop damping materials. Among them, polymer-based damping materials attract the most attention due to their unique viscoelastic properties [4,5,6]. For traditional polymer-based damping materials, such as blending [7], copolymerization [8] and interpenetrating network materials [9], external mechanical energy dissipated through the internal friction of molecular chains is a major step towards improvement of damping properties [10,11,12]. However, such energy dissipation may find it hard to satisfy the increasing requirement today [13]. Therefore, the development of polymer-based materials with new energy dissipation is imperative.

In recent years, the use of dynamic noncovalent bonds to enhance energy dissipation is proposed [14,15,16,17]. Compared with the traditional internal friction mechanism, the reversible destruction and re-construction of noncovalent bonds additionally convert external mechanical energy into thermal energy. Thus, the damping properties are remarkably improved [18,19,20,21,22,23]. So far, this energy dissipation mechanism mainly exists in hindered phenol/polymer-based hybrid damping systems [19,20,21,22,23,24,25,26,27,28,29,30,31,32,33,34,35,36,37]. Hindered phenols, such as AO-60 [19], AO-70 [20,21] and AO-80 [22,23,24], are added into polar polymers to form intermolecular hydrogen bonds (HBs). The existence of the intermolecular HBs is qualitatively verified by means of infrared spectroscopy [28], temperature-dependent infrared spectroscopy [23,25], two-dimensional infrared spectroscopy [20,21] and nuclear magnetic resonance [19,25]. Moreover, to reveal the quantitative relationship between intermolecular HBs and damping properties of the hybrids, molecular dynamics (MD) simulation, with the ability to predict material properties at the atomic or molecular level, is employed [19,20,21,23,24,25,29]. The simulation results show that the damping property of the hybrids is dominated by the number of intermolecular HBs and the binding energy, as well as the fractional free volume (FFV). Therefore, the binding energy and FFV are mainly affected by the intermolecular HBs. A sufficient number of intermolecular HBs are thus required for high-performance hybrid damping materials. However, due to the high cohesive energy density, the intramolecular HBs between hindered phenols compete with the intermolecular HBs at room temperature, which leads to easy self-aggregation and crystallization of hindered phenols in long-term service. As a result, the damping property dramatically decreases with the decreasing number of intermolecular HBs [30,31,32,33,34,35].

Therefore, design and optimization of the molecular structure of hindered phenols are necessary to improve dispersion stability in polymer matrices and are key issues in the practical application of hybrid damping materials [37]. To our surprise, the design and optimization of the molecular structure have rarely been reported. Only a few researchers propose the method of increasing the volume of hindered phenols to improve dispersion stability. For example, by constructing bulk-like structures, such as grafting hindered phenol to polymer chains [38,39] or to the surface of inorganic nanoparticles [40] and by synthesizing hyperbranched hindered phenol molecules [41], the intramolecular HBs are remarkably weakened. Thus, the self-aggregation and crystallization are successfully restricted. However, the damping property of the hybrids shows little improvement because of the simultaneous steric effects on the intermolecular HBs [42]. Therefore, a new optimization method of hindered phenol for high damping performance is urgently required.

As mentioned above, the self-aggregation and crystallization of hindered phenol are mainly attributed to the high cohesive energy density. The cohesive energy density is mainly affected by molecular polarity; high polarity leads to high cohesive energy density. Relating to the molecular structure of the commonly used hindered phenols, such as AO-60, AO-70 and AO-80, the high cohesive energy density may be attributed to the high density of polar groups in short linear backbones. Thus, the decrease of the density of polar groups through the increase of the nonpolar backbone length may lead to a stable hindered phenol. Moreover, as the end groups in the linear chain, the intermolecular HBs between hydroxyl groups in hindered phenol and polar groups in polymer may be not hindered. Therefore, the aim was to design a stable and high-performance hindered phenol/polymer-based damping material, and the molecular structure of a linear hindered phenol was optimized through the construction of long nonpolar backbone length for the first time, in this study. The damping properties of the hindered phenol/TPU hybrids were evaluated by a combination of MD simulation and dynamic mechanical analysis. The purpose of this study is to: (1) design and optimization of the molecular structure of hindered phenol to improve dispersion stability and damping performance and (2) explore the relationship between structure optimization and damping property.

## 2. Experimental Section

### 2.1. Materials

Polyester-based TPU (Elastollan S85A11), with hardness 85 shore A and 4,4’-methane diisocyanate (MDI) within the hard segments, was purchased from Elastogran, BASF group (Lemförde, Germany). Hindered phenol, triethylene glycol bis(3-tertbutyl-4-hydroxy-5-methylphenyl)propionate (AO-70), in powder form, was obtained from Beijing Additive Research Institute (Beijing, China). 1,12-Dodecanediol and ethyl acetate were purchased from Aladdin Industrial Corporation (Industry, CA, USA). Hydrochloric acid, sodium hydroxide, dichloromethane, cyclohexane, sodium bicarbonate and p-toluene sulfonic acid were obtained from Chengdu Kelong Chemical Reagent Factory (Chengdu, China). All the materials were used without further purification.

### 2.2. Synthesis of the Linear Hindered Phenol: Dodecane-1,12-diyl bis 3-(3-tertbutyl-4-hydroxy-5-methylphenyl)propionate (DP)

(1) Synthesis of 3-(3-tertbutyl-4-hydroxy-5-methylphenyl)propionic acid (AO): In a one neck round bottom flask, 96 g (2.4 mol) sodium hydroxide was dissolved in 800 mL of distilled water, then 176.04 g (0.3 mol) AO-70 was added under stirring. The mixture was stirred at 90 °C with a reflux condenser for 8 h under nitrogen atmosphere. After the reaction, the aqueous phase was extracted with dichloromethane four times, then was acidized by diluted hydrochloric acid to pH 3.0. White sediment appeared in the process of hydrochloric acid titration. After stewing for 12 h, the sediment was filtrated and washed to pH 7.0 with distilled water. The resulting white powder was dried in an air-circulating oven at 80 °C for at least 12 h before use.

(2) Synthesis of DP: In a one neck round bottom flask with 300 mL cyclohexane, 13.13 g (0.065 mol) 1,12-Dodecanediol, 61.36 g (0.26 mol) AO and 0.6565 g p-toluene sulfonic acid were added under stirring. The mixture was stirred at 95 °C with a water separator for 8 h under nitrogen atmosphere. After the reaction, the solution was titrated with saturated sodium bicarbonate solution to pH 8.0, then was extracted with ethyl acetate. The organic layer was retained and washed to pH 7.0 with distilled water. After the evaporation of ethyl acetate in the organic layer by rotary evaporation, the remaining transparency liquid was dried in an air-circulating oven at 80 °C for 12 h. DP with relative molecular weight of 638 and transparency solid state was then obtained.

### 2.3. Preparation of TPU/DP Hybrids

The binary hybrids were prepared as follows: (1) The as-received TPU and DP were dried in a vacuum oven at 80 °C and 60 °C for 12 h. (2) 50 g dried TPU was first hot sheared in a RM-200C torque rheometer (HAPPO, China) at a rotor speed of 30 rpm for 2 min at 185 °C. (3) DP with weight of 0, 2.5, 7.5, 12.5 and 17.5 g (i.e., weight ratio of 0, 5, 15, 25 and 35 phr) were added to the sheared TPU, respectively. (4) The mixtures were mixed in the torque rheometer at a rotor speed of 30 rpm for another 10 min at 185 °C to prepare the binary hybrids. These are designated, respectively, as TDP0 to TDP35.

In order to obtain mixed samples for characterization, the hybrids were first dried at 80 °C for 12 h, then preheated at 185 °C for 8 min, hot-pressed at 185 °C for 4 min under 12 MPa, and then cool-pressed at room temperature under 12 MPa.

### 2.4. Characterization

Nuclear magnetic resonance spectroscopy (^1^H NMR) were recorded on a Bruker Avance III 400 spectrometer (Bruker Daltonics Inc., Karlsruhe, Germany) at room temperature using CDCl_3_ as solvent. Fourier transform infrared spectoscopy (FT-IR) data of AO and DP were obtained with KBr Pellets on a PerkinElmer Spectrum Two instrument (PerkinElmer, Waltham, MA, USA), utilizing a range of 4000 to 400 cm^−1^ with a resolution of 2 cm^−1^, and FT-IR data of the hybrids were obtained using the attenuated total reflectance (ATR) accessory (PerkinElmer, Waltham, MA, USA). The X-ray diffraction (XRD) patterns of the samples were measured on a Tongda TD-3500X X-ray diffractometer (Dandong Tongda Instrument Co., Ltd., Dandong, China) using Cu/Ka radiation (λ = 0.154056 nm) in the 2θ range from 5° to 45°. Dynamic mechanical analysis (DMA) were acquired using a PerkinElmer Q8000 dynamic mechanical analyzer (PerkinElmer, Waltham, MA, USA) at a fixed frequency of 1 Hz. DMA analyses were performed in double cantilever mode, using specimens with approximate dimensions of 8 mm (length) × 10 mm (width) × 1 mm (thickness). The samples were heated at temperatures ranging from −60 to 80 °C at a heating rate of 3 °C/min.

### 2.5. MD Simulation for TPU/DP Hybrids

MD simulation was performed by using Amorphous cell, Discover and Forcite modules (Accelrys, Inc., San Diego, CA, USA) in Material Studio Modeling software (version 7.0) [43]. The condensed-phase optimized molecular potentials for atomistic simulation studies (COMPASS) forcefield, which had been widely used to optimize and predict the structural, conformational, and thermophysical condensed phase properties of polymers [44], was used in this study. The temperature and pressure regulation depended on the Andersen and Berendsen method [45]. The non-bonded interaction and van der Waals interactions were determined by standard atom-based simulation method [46].

The procedures of MD simulation are shown in Figure 1. Based on our previous study [20] and the practical structure used in this study, the hard segment (Figure 1a) and soft segment (Figure 1b) repeating units were firstly predigested constructed. Then, a TPU polymer chain consisting of 18 hard and soft repeating units was built with five degrees of polymerization adopted (Figure 1c). Secondly, the amorphous cell containing one energy optimized TPU chain (Figure 1c) and a certain amount of DP molecule (Figure 1d) was constructed with a periodic boundary (Figure 1e) adopted. The cell was firstly energy minimized by means of the steepest descent and conjugate gradient method. Subsequently, the cell was equilibrated in the isothermal–isobaric (NPT) ensemble at 1 atm and 25 °C to obtain an energy equilibrated cell. The structural relaxation process was done for 1 nanosecond (ns) with the dynamics that was followed by data accumulation running for another 1 ns and the configurations saved for every 5 picoseconds (ps) (Figure 1f). Finally, the cell was used for counting useful information, such as the number of HBs (Figure 1g) and FFV (Figure 1h).

## 3. Results and Discussion

### 3.1. Structure of AO and DP

To confirm the structure of AO, ^1^H NMR and FT-IR were adopted. For the ^1^H NMR result in Figure 2a, the double peaks observed at about 6.973 ppm and the singlet at 6.848 ppm are attributed to Ph-H. The singlet at 4.662 ppm belongs to Ph-OH. The multiple peaks observed between 2.839–2.879 ppm and 2.623–2.662 ppm are attributed to Ph-CH_2_-CH_2_-. The singlets at 2.225 ppm and 1.399 ppm belong to Ph-CH3 and Ph-tBu, respectively [42,47,48]. For FT-IR result in Figure 2b, the peak at 3543.52 cm^−1^ is attributed to the hindered hydroxyl group and the peaks at about 3200–3000 cm^−1^ and 1694.15 cm^−1^ are attributed to the carboxyl group [42,47]. Hence, ^1^H NMR combined with FT-IR clearly confirms the structure and functional groups of AO.

The structure and micromorphology of DP were examined by ^1^H NMR, FT-IR and XRD. The results are shown in Figure 3. For the ^1^H NMR result in Figure 3a, though peaks at 6.944, 6.803, 5.103, 2.817–2.855, 2.559–2.597, 2.169 and 1.393 ppm show a shift compared with that of AO monomer, the contributors are the same. Besides, the multiple peaks observed between 4.041–4.074 ppm are attributed to 1- (or 12-) methylene of 1,12-Dodecanediol monomer, and the singlets at 1.590 and 1.264 ppm are attributed to 2- (or 11-) methylene and 3- to 6- (or 7- to 10-) methylene of 1,12-Dodecanediol monomer, respectively [49,50]. For the FT-IR result in Figure 3b, the peaks observed at 3495.30 and 1717.56 cm^−1^ are attributed to the hydroxyl and carbonyl group, respectively, and the peaks observed at 1054.05 and 1028.08 cm^−1^ are attributed to the ether linkage [51]. Hence, the successful synthesis of DP is confirmed. For the XRD result in Figure 3c, only a 2θ at about 20° can be observed, indicating that noncrystalline (or amorphous) hindered phenol is obtained [52].

### 3.2. The Character of TDP Hybrids

To preliminary study the HBs and micromorphology of TDP hybrids, FT-IR and XRD were adopted. In Figure 4a, by increasing the amount of DP, an IR peak at about 3480 cm^−1^, attributed to the hydrogen bonded hydroxyl group, gradually appears; the peak at about 3327 cm^−1^ related to the amino group in TPU shows a blue shift while the peak at about 1728 cm^−1^ related to the carbonyl group in TPU shows a red shift [20]. The blue shift indicates the weakness of the intramolecular HBs between amino and carbonyl group in TPU, while the red shift indicates the enhancement of the intermolecular HBs between the hydroxyl group in DP and the carbonyl group in TPU. In Figure 4b, the XRD curves exhibit a single 2θ peak at about 21°, which indicates the amorphous state of all hybrids [53].

### 3.3. MD Simulation of TDP Hybrids

For hindered phenol/polymer-based hybrid damping materials, the reversible destruction and re-construction of intermolecular HBs combined with the internal friction of molecular chains are the determining factors for damping performance. To explore the two factors of the amorphous hindered phenol/TPU hybrids in a quantitative manner, MD simulation, based on an amorphous cell, was further adopted.

Related to the chemical structure of DP and TPU, two types of proton donators (i.e., urethane N-H group in the hard segment of TPU (1) and phenolic O-H group in DP (2)) and three types of proton acceptors (i.e., urethane C=O group in the hard segment of TPU (3), ester C=O group in the soft segment of TPU (4) and ester C=O group in DP (5)) exist. Thus, the intermolecular HBs within 1-5, 2-3, 2-4 and the intramolecular HBs within 1-3, 1-4, 2-5 may be formed. Therefore, the molar concentration of the abovementioned HBs was explored; the results are shown in Figure 5. The molar concentration of HBs (*C*_HBs_) is calculated using the following equation:(1)CHBs=NHBsNAV
where *N*_HBs_, *NA* and *V* are the statistical average number of HBs in the periodic cell [29], Avogadro’s constant and the volume of the periodic cell, respectively. By increasing the amount of DP, the *C*_HBs_ between 2 and 4 gradually increases from 0 to 24.5, indicating that more and more intermolecular HBs between 2 and 4 are formed. The *C*_HBs_ = 0 for TDP5 may be due to the long backbone length of DP. The *C*_HBs_ between 1 and 4 gradually decreases from 34 to 24, indicating that the intramolecular HBs between 1 and 4 gradually decrease. While the *C*_HBs_ between 1 and 3 fluctuates in a small range, indicating that the intramolecular HBs between 1 and 3 change a little. The above simulation results are in accordance with the FT-IR results, and the blue shift of the amino group can therefore be attributed to the decrease of the intramolecular HBs between ester C=O group and urethane N-H group in TPU. Besides, the intermolecular HBs within 1-5, 2-3 and the intramolecular HBs within 2-5 were not detected for all TDP hybrids; therefore, the results are not depicted here.

For the damping property of TDP hybrids, the strength of the intermolecular HBs are also a key factor [54]. Therefore, the strength of the intermolecular HBs between phenolic O-H group in DP and ester C=O group in the soft segment of TPU was also explored. The strength of HBs can be indirectly judged through geometric parameters: bond angle and length. Generally speaking, a larger bond angle and a smaller bond length correspond to stronger HBs [25]. For TDP hybrids, the average angles of intermolecular HBs in TDP15, TDP25 and TDP35 systems are 136.931°, 138.255° and 139.826°, with the average lengths of 2.216, 2.189 and 2.175 Å, respectively. Therefore, the relative intermolecular HBs strength is TDP35 > TDP25 > TDP15.

The binding energy (*E*_binding_), which is defined as the negative of the intermolecular interaction energy (*E*_inter_) between two components, was introduced to further study the TDP hybrids [55]. The *E*_inter_ can be evaluated by the total energy (*E*_total_) of the hybrid and those of the individual components in the equilibrium state. Thus, the *E*_binding_ of TDP hybrids can be determined as follows:(2)$$E_{\mathrm{binding}}=-E_{\mathrm{inter}}=-\left(E_{\mathrm{total}}-E_{\mathrm{DP}}-E_{\mathrm{TPU}}\right)$$
where *E*_total_, *E*_DP_ and *E*_TPU_ are the total energy of the TDP hybrid, DP and TPU, respectively. The *E*_TPU_ is a constant (−2985.37 kcal mol^−1^) because of the fixed number of TPU chain in the amorphous cell. Table 1 displays the binding energies of TDP hybrids. Negative *E*_total_ values indicate that the TDP hybrids were stable [20,23]. For TDP5, the binding energy was the lowest because of the absence of intermolecular HBs. By increasing the amount of DP, the binding energy shows a big increase, which is in accordance with the variation of the intermolecular HBs concentration and strength. Thus, the existence of the intermolecular HBs leads to the enhancement of the intermolecular interaction energies.

The internal friction of molecular chains is mainly influenced by chain packing. To characterize the efficiency of chain packing, FFV was adopted. A common definition of FFV is:(3)$$\mathrm{FFV}=\frac{V-V^{*}}{V}$$
where *V* and *V** are the specific volume and occupied volume, respectively [56]. In Figure 6a, by increasing the amount of DP, a decrease in FFV can be obviously observed, which indicates that the chain packing between DP and TPU becomes more and more compact. Compared with the low content systems, the FFV shows a drastic decreasing degree from TDP25 to TDP35. To explore the variation, the respective chain packing of TPU and DP was studied with the help of Radius of gyration (Rg) and the relative concentration of segments. A large Rg indicates an extended chain packing and a small Rg indicates a compact chain packing. In Figure 6b, the Rg of TPU chain increases first and then decreases; TDP35 exhibits the lowest value compared with other TDP hybrids. The Rg increase for TDP5 and TDP15 may because of the relatively low intermolecular interactions and the Rg decrease for TDP25 and TDP35 because of the high intermolecular interactions. In Figure 6c, the fluctuation of the relative concentration of TPU chain increases with increasing the amount of DP. The maximum value appears in TDP35. In relation to the intramolecular HBs variations in TPU, the variation of the relative concentration may be because of the separation effects of DP to the soft and hard segments of TPU. In Figure 6d, the Rg of DP chain increases at different rates, which indicates that the linear DP chains become more and more extended in the soft segments of TPU. Thus, because the TPU chain becomes more compact in high DP content systems, the DP chain becomes more stable on increasing the amount of DP, which is benefit for polymer-based damping systems.

### 3.4. Damping Property and Mechanism of TDP Hybrids

To explore the damping mechanism of TDP hybrids, the damping properties were explored; the results are depicted in Figure 7 and Table 2. For TDP hybrids, only one tanδ peak, attributed to the glass transition of TPU, is observed in the whole temperature range, indicating that TPU and DP are compatible. By increasing the amount of DP, the peak position gradually shifts to a higher temperature, while the peak value decreases first and then increases. It should be noted that: (1) the decrease of the peak value is observed for the first time in linear hindered phenol/polymer-based hybrids and (2) the peak value shows a large increase for TDP35. Moreover, the TA value shows a similar trend with that of tanδ, while the variation of ΔT is different. The ΔT value for TDP15 and TDP25 are larger than that for TDP0, while the value for TDP5 and TDP35 are smaller than that for TDP0. In Figure 7b, the variation of the storage modulus is in accordance with the variation of tanδ.

Combined with the MD simulation results above, the damping mechanism of TDP hybrids can be concluded as follows: Because of the long and low polar chain structure, intermolecular HBs between DP and TPU in TDP5 cannot be formed. DP only acts as a plasticizer in TPU matrix, which leads to the extended chain packing of TPU and results in the decrease of the damping property. In TDP15, the intermolecular HBs are formed with low bond concentration and strength, which have little effects on the chain packing of TPU; the main energy dissipation is through intermolecular interactions. Because of the low value of the binding energy, the damping property of TDP15 increases a little. With the bond concentration and strength of the intermolecular HBs further increasing, the chain packing of TPU becomes more compact and the DP chains become more extended in the soft segments of TPU. Thus, the intermolecular interactions and chain friction are increased, which lead to the improvement of damping property for TDP25. For TDP35, the largest bond concentration and strength of the intermolecular HBs lead to the strongest intermolecular interactions, which results in the most compact chain packing of TPU and the most extended chain packing of DP as well as the closest contact between TPU and DP chains. The energy dissipates through the synergistic effects of the reversible destruction, re-construction of intermolecular HBs, and internal friction of chains. Thus, the damping property shows remarkable improvement. Besides, it should be noted that the strongest intermolecular interactions and the closest chain packing also restrict chain relaxation within a narrow temperature range.

## 4. Conclusions

In this study, a linear hindered phenol with amorphous state and low polarity was synthesized, for the first time, to improve the stability and damping performance of hindered phenol/polymer-based damping materials. The damping mechanism of the amorphous hindered phenol/TPU hybrid systems was thus revealed. The conclusions are as follows:

(1) By simply increasing the chain length to decrease the chain polarity of a linear crystalline hindered phenol, the intramolecular HBs and cohesive energy density are restricted, and an amorphous hindered phenol is synthesized.

(2) Stable hindered phenol/polymer-based hybrid damping materials is obtained by introduction of the synthetic hindered phenol. Because of the plasticizing effects of hindered phenol in TDP5, the damping property of the hybrid shows a decrease. In high DP content systems, stable and high-performance hybrid damping materials are obtained.

(3) For hybrid damping materials, the chain polarity and chain size of hindered phenol have significant effects on hydrogen bonding interactions. The concentration and strength of the intermolecular HBs dominate the chain packing of the hybrids, which then affect the damping performance. A large bond concentration and strength lead to compact chain packing of the polymer, extended chain packing of hindered phenol, and close contact between hindered phenol and polymer. Thus, the synergistic effect of the intermolecular interactions and the internal friction of the chains results in good damping performance. However, the effects of the small bond concentration and strength are opposite. Thus, only energy dissipation through intermolecular interactions results in poor damping performance.

## Figures and Tables

**Figure 1 polymers-11-00884-f001:**
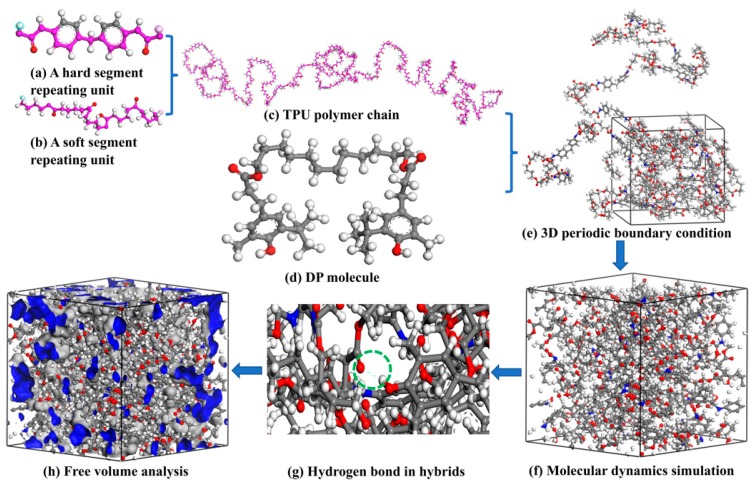
Models for MD simulation of TPU/DP hybrids (grey sphere represents C atom; white sphere represents H atom; red sphere represents O atom; blue sphere represents N atom and blue dashed line represents HBs): (**a**) hard segment repeating unit; (**b**) soft segment repeating unit; (**c**) TPU polymer chain; (**d**) DP molecule; (**e**) amorphous cell with a periodic boundary adopted; (**f**) energy minimization of amorphous cell; (**g**) hydrogen bonds calculation and (**h**) free volume analysis.

**Figure 2 polymers-11-00884-f002:**
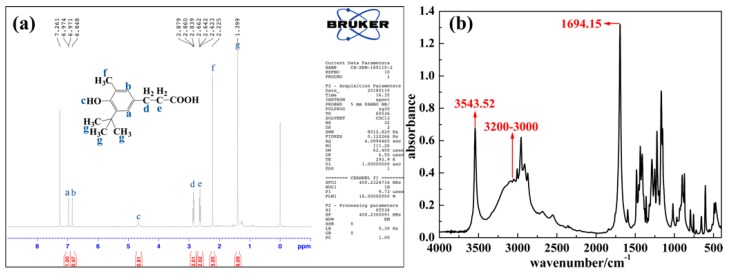
^1^H NMR spectra (**a**) and FT-IR spectra (**b**) of AO.

**Figure 3 polymers-11-00884-f003:**
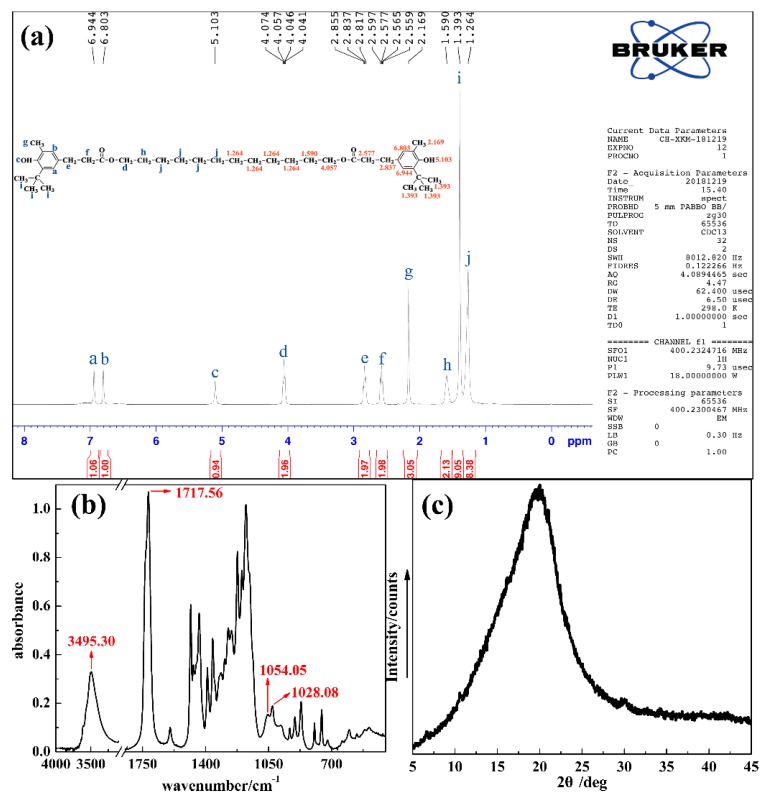
^1^H NMR spectra (**a**), FT-IR spectra (**b**) and XRD curve (**c**) of DP.

**Figure 4 polymers-11-00884-f004:**
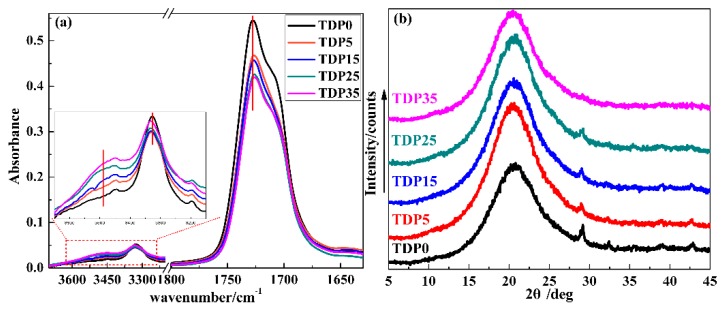
FT-IR spectra (**a**) and XRD curves (**b**) of TDP hybrids.

**Figure 5 polymers-11-00884-f005:**
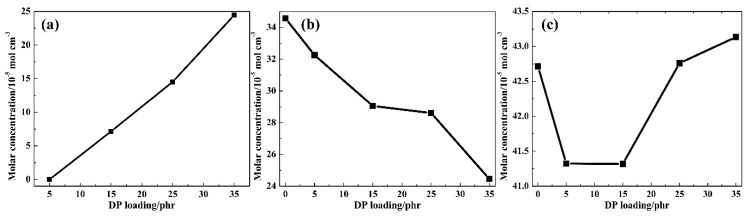
Intermolecular *C*_HBs_ between phenolic O-H group in DP and ester C=O group in the soft segment of TPU (**a**); intramolecular *C*_HBs_ between urethane N-H group in the hard segment of TPU and ester C=O group in the soft segment of TPU (**b**) and intramolecular *C*_HBs_ between urethane N-H group in the hard segment of TPU and urethane C=O group in the hard segment of TPU (**c**) of TDP hybrids at 298 K.

**Figure 6 polymers-11-00884-f006:**
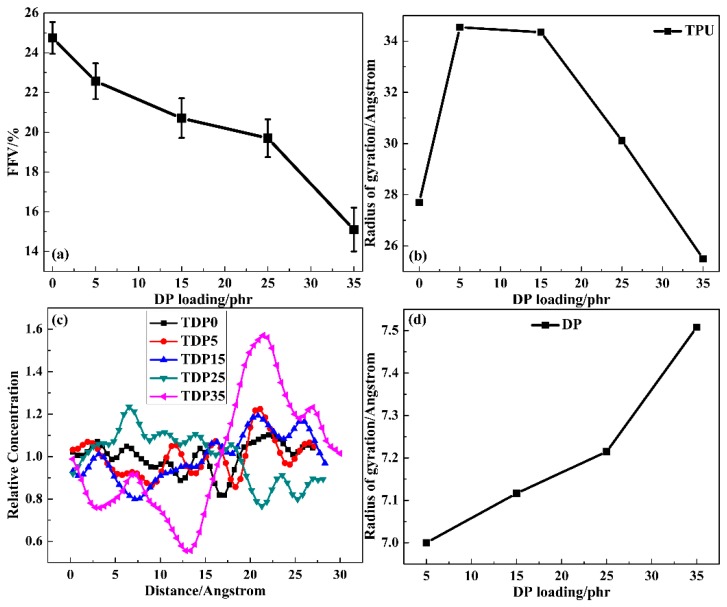
Fractional free volume of TDP hybrids (**a**); Radius of gyration for TPU chain with different DP loading (**b**); relative concentration of TPU chain along X axis (**c**) and radius of gyration for DP chain (**d**) with different DP loading.

**Figure 7 polymers-11-00884-f007:**
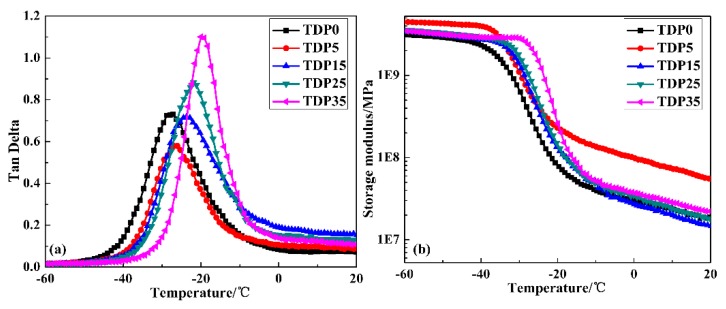
Temperature dependence of (**a**) loss factor (tanδ) and (**b**) storage modulus of TDP hybrids.

**Table 1 polymers-11-00884-t001:** Binding energies of TDP hybrids with different numbers of DP.

Sample Name	*E*_total_/kcal mol^−1^	*E*_DP_/kcal mol^−1^	*E*_binding_/kcal mol^−1^
**TDP5**	−3260.71	−255.07	20.27
**TDP15**	−3532.09	−461.37	86.09
**TDP25**	−3748.53	−607.34	155.82
**TDP35**	−4096.18	−821.52	289.29

**Table 2 polymers-11-00884-t002:** Damping parameters of TDP hybrids with different loadings of DP.

Sample Name	Tanδ_max_		Tanδ > 0.3			
	T_g_/°C	Value	T_begin_	T_end_	ΔT/°C	TA
**TDP0**	−27.14	0.73	−35.65	−17.02	18.63	10.22
**TDP5**	−26.11	0.58	−32.91	−18.55	14.36	7.24
**TDP15**	−23.74	0.72	−31.90	−10.88	21.02	11.63
**TDP25**	−21.90	0.88	−30.24	−10.21	20.03	12.17
**TDP35**	−19.98	1.10	−26.65	−9.79	16.86	12.35

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
