# Peer review of "Towards a Stable and High-Performance Hindered Phenol/Polymer-Based Damping Material Through Structure Optimization and Damping Mechanism Revelation"

_polymers, 2019, doi:10.3390/polym11050884_

Round 1
Reviewer 1 Report
This manuscript studies the damping properties of hindered phenol/polymer-based material and its damnping mechanism. Overall the manuscript fails to show scientific rigor in directly demonstrating the revelation between molecular structures and damping mechanism. It is recommended that manuscript require major revision before possible publication. Review comments are as follows.
The molecular dynamics data presented in the manuscript are insufficient to support the arguments on the hydrogen bonds and their effects on the damping. Please provide N_{HBs} data from the MD calculations.
Equations 2 and 3 are incorrectly presented.
Please scientifically demonstrate the 'reversible destruction and re-construction of intermolecular HBs' in the TDP35 sample.
If the 'reversible destruction and re-construction of intermolecular HBs' is the correct additional damping mechanism (additional to the internal friction), why can this mechanism provide such large increase in damping (over 20%)? Please provide quantitative calculations.
Bond breaking can lead to increases in energy dissipation, i.e. damping. However, bond formation consumes energy, in general. Hence, explanations are in order to support the proposed mechanism.
What would happen for TDP40 or TDP45 samples? More damping enhancements?
Are the damping peaks in Figure 7 (a) alpha, beta or gamma peaks?
In Figure 7 (b), why TDP35 sample shows a slight increase in storage modulus around -28 degC before the transition?
English requires significant editing. Lines 35-37 need to be re-written. In Lines 39-41, new energy dissipation mechanisms are mentioned. Please provide examples to delineate the new mechanisms that have been published in the literature.
Author Response
Response to Reviewer 1 Comments
This manuscript studies the damping properties of hindered phenol/polymer-based material and its damping mechanism. Overall the manuscript fails to show scientific rigor in directly demonstrating the revelation between molecular structures and damping mechanism. It is recommended that manuscript require major revision before possible publication. Review comments are as follows.
Point 1: The molecular dynamics data presented in the manuscript are insufficient to support the arguments on the hydrogen bonds and their effects on the damping. Please provide N_{HBs} data from the MD calculations.
Response 1: Thank you for your suggestion. The corresponding revision has been done in revised manuscript. The simulate hydrogen bonds between N-H and C=O have been added into Fig. 5, and the related explains have also been given.
Point 2:Equations 2 and 3 are incorrectly presented.
Response 2:Thank you for your suggestion. The corresponding revision has been done in revised manuscript.Equations 1, 2 and 3 have been replaced with an image format.
Point 3:Please scientifically demonstrate the 'reversible destruction and re-construction of intermolecular HBs' in the TDP35 sample.
Response 3:Thank you for your suggestion. The corresponding revision has been done in revised manuscript.Essentially,the reversible destruction and re-construction of the intermolecular HBs in TDP35 are the same as other TDP hybrids. The difference between them are: (1) the bond concentration and strength of TDP35 are the largest, which lead to the largest intermolecular interactions. Thus, (2) the most compact chain packing of TPU and the most extended chain packing of DP as well as the closest contact between TPU and DP chains are existed in TDP35. Therefore, the synergistic effects of the reversible destruction and re-construction of intermolecular HBs and the internal friction of chainsresult in the great improvement of the damping property. The above demonstration has been added in 3.4 part.
Point 4:If the 'reversible destruction and re-constructionof intermolecular HBs' is the correct additional damping mechanism (additional to the internal friction), why can this mechanism provide suchlarge increase in damping (over 20%)? Please provide quantitative calculations.
Response 4:Thank you for your suggestion. The corresponding revision has been done in revised manuscript. Zhu and co-workers have revealed the quantitative relationship. They establish the linear correlation between the energy dissipation parameter at certain temperature (μT) of hydrogen bond dissociation reactionand the tanδmaxwith the R2≥0.924. The μTis defined as:
, where cis constant, T0refers to the temperature at which ΔG?is equal to 0 kJ mol-1, K?refers to the equilibrium constant of the hydrogen bond dissociation reaction, Tis reaction temperature and ∆??????????reflects the heat change of the reaction under the special condition of constant pressure. We agree with them and have cited the literature in our manuscript. Moreover, the large increase in damping (over 20%) for TDP system is not only attributed to the intermolecular HBs, but also attributed to the enhanced internal friction.
Point 5:Bond breaking can lead to increases in energy dissipation, i.e. damping. However, bond formation consumes energy, in general. Hence, explanations are in order to support the proposed mechanism.
Response 5:Thank you for your suggestion. The corresponding revision has been done in revised manuscript.The HBs in TDP system is naturally stable, the breaking of them is an endothermic reaction, which consumes energy. During dynamic processes, the external mechanical energy provides the energy for the breaking. As an equilibrium reaction, the re-formation of the HBs is an exothermic reaction, which releases thermal energy and occurs spontaneously. By the breaking and the re-formation of the HBs, the external mechanical energy is ultimately converted into thermal energy, which increases the energy dissipation.
Point 6:What would happen for TDP40 or TDP45 samples? More damping enhancements?
Response 6:Thank you for your suggestion. The corresponding revision has been done in revised manuscript.There are two types of proton acceptors (urethane C=O and ester C=Ogroups) in TPU. For TDP0 to TDP35, only the intermolecular HBs between ester C=O group in TPU and O-H group in DP are formed, which leads to the increase of damping property. According to our previous study (DOI: 10.1039/C4TA00476K), with the amount of DP further increasing, if the intermolecular HBs between ester C=O group in TPU and O-H group decrease and convert into the intermolecular HBs between urethane C=O in TPU and O-H group, the damping property decreases; if the above two intermolecular HBs both increase and form an intermolecular HBs network, the damping property increases. For most polymer matrix, there is only one type of proton acceptor, no intermolecular HBs transform occurs. Therefore, we only discussed the TDP0 to TDP35 system with one type of intermolecular HBs in this paper.
Point 7:Are the damping peaks in Figure 7 (a) alpha, beta or gamma peaks?
Response 7:Thank you for your suggestion. The corresponding revision has been done in revised manuscript.The damping peaks in Figure 7 (a) are attributed to the glass transition of the soft segments of TPU, therefore, the damping peaks are alpha peaks.
Point 8:In Figure 7 (b), why TDP35 sample shows a slight increase in storage modulus around -28 degC before the transition?
Response 8:Thank you for your suggestion. The corresponding revision has been done in revised manuscript.The slight increase may because of the cold crystallization of the hard segments of TPU.
Point 9:English requires significant editing. Lines 35-37need to be re-written. In Lines 39-41, new energy dissipation mechanisms are mentioned. Please provide examples to delineate the new mechanisms that have been published in the literature.
Response 9:Thank you for your suggestion. The corresponding revision has been done in revised manuscript. English has been re-edited.Lines 35-37 have been re-written. And examples have been added for the new energy dissipation mechanisms.

Reviewer 2 Report
The idea and experimental data of work are interesting. However, a significant revision is required before it is accepted to publish in this journal. My comments are as follows:
1. Abstract needs to be thoroughly revised, more experimental data should be added.
2. Introduction, the importance of this work to other scholars and reader should be highlighted.
3. Section 3.1. Please give more relevant citations when the authors discussed the experimental data of NMR, FT-IR, and XRD.
4. Please give more relevant citations when the authors discussed the experimental data in Section Results and discussions.
5. Please not that any data reused (i.e., DOI: 10.1039/C4TA00476K) must be appropriately citated. Otherwise, it will be self-plagiarism.
6. A similarity report of revised manuscript needs to be attached.
7. English needs to be checked by a native speaker, too many grammar mistakes were found.
8. The resolution of most figures needs to be increased (i.e., the DPI resolution should be higher than 600)
9. too many format errors, please revise carefully. For example:
+ “PH=3” should be pH 3.0”, please revise for the same mistakes.
+ “berendsen method[42]”…… should be “berendsen method [42].”, please revise for the same mistakes.
Author Response
Response to Reviewer 2 Comments
The idea and experimental data of work are interesting. However, a significant revision is required before it is accepted to publish in this journal. My comments are as follows:
Point 1: Abstract needs to be thoroughly revised, more experimental data should be added.
Response 1: Thank you for your suggestion. The corresponding revision has been done in revised manuscript. The abstract is thoroughly revised and more experimental data are added.
Point 2:Introduction, the importance of this work to other scholars and reader should be highlighted.
Response 2:Thank you for your suggestion. The corresponding revision has been done in revised manuscript.The importance of this work has been highlighted.
Point 3:Section 3.1. Please give more relevant citations when the authors discussed the experimental data of NMR, FT-IR, and XRD.
Response 3:Thank you for your suggestion. The corresponding revision has been done in revised manuscript.More relevant citations have been added.
Point 4:Please give more relevant citations when the authors discussed the experimental data in Section Results and discussions.
Response 4:Thank you for your suggestion. The corresponding revision has been done in revised manuscript. More relevant citations have been added.
Point 5:Please not that any data reused (i.e., DOI: 10.1039/C4TA00476K) must be appropriately citied. Otherwise, it will be self-plagiarism.
Response 5:Thank you for your suggestion. The corresponding revision has been done in revised manuscript. The article (DOI: 10.1039/C4TA00476K) has been citied.
Point 6:A similarity report of revised manuscript needs to be attached.
Response 6:Thank you for your suggestion. In the revised manuscript, the revised part has been written in red, which may be helpful for your vetting.
Point 7:English needs to be checked by a native speaker, too many grammar mistakes were found.
Response 7:Thank you for your suggestion. The corresponding revision has been done in revised manuscript.
Point 8:The resolution of most figures needs to be increased (i.e., the DPI resolution should be higher than 600)
Response 8:Thank you for your suggestion. The corresponding revision has been done in revised manuscript.The resolution of all figures has been increased to 1200 DPI.
Point 9:too many format errors, please revise carefully. For example:
+ “PH=3” should be pH 3.0”, please revise for the same mistakes.
+ “berendsen method[42]”…… should be “berendsen method [42].”, please revise for the same mistakes.
Response 9:Thank you for your suggestion. The corresponding revision has been done in revised manuscript.

Reviewer 3 Report
The intend of authors is to provide both experimental and simulation tests of the hindered Phenol/polymer-based damping material, in which the topic is interesting and important form the application point of view. However there is lack of clarity in the discussion, and another revision is required form my side.
1. On page 4, the authors gave a flow chart of the preparation of the TPU/DP hybrid materials. However, for TPU, the composition ratio of the hard and soft segment is not mentioned in the recent work. Similarly, the authors have not presented the ratio between TPU and DP. The readers would be confused if the molecular weight of TPU is not significantly greater than that of DP, the system volume will change greatly with different filling density of DP, and further influence the results of FFV as well as the molar concentration of HBs.
2. On page 4, the authors mentioned that the periodic boundary condition is adopted in the current work, so why the “free volume” appears to distribute on the inner surface of the cell, according to Fig. 1(h)?
3. On page 5, for the FTIR figure, Why the authors consider the peak appears at 3143.61, instead of 3000? Besides, I think the references are necessary for the identification of those peaks.
4. In Fig. 6(a), I recommend the authors to add the error bar and/or present another work at different DP loading, to get the conclusion of “the FFV showed a drastic decreasing degree from TDP25 to TDP35 compared with that for the low content systems”. In addition, the results of Fig 6(b) and (c) are quite interesting and obviously hard to understand for most readers, so I recommend the authors to explain the underlying reasons, instead of reporting them.
Author Response
Response to Reviewer 3 Comments
The intend of authors is to provide both experimental and simulation tests of the hindered Phenol/polymer-based damping material, in which the topic is interesting and important form the application point of view. However there is lack of clarity in the discussion, and another revision is required form my side.
Point 1: On page 4, the authors gave a flow chart of the preparation of the TPU/DP hybrid materials. However, for TPU, the composition ratio of the hard and soft segment is not mentioned in the recent work. Similarly, the authors have not presented the ratio between TPU and DP. The readers would be confused if the molecular weight of TPU is not significantly greater than that of DP, the system volume will change greatly with different filling density of DP, and further influence the results of FFV as well as the molar concentration of HBs.
Response 1: Thank you for your suggestion. The corresponding revision has been done in revised manuscript. About the composition ratio of the hard and soft segment of TPU, the information cannot be obtained from BASF company. But the hardness can indirectly reflect the ratio. A high hardness indicates a high ratio of the hard segment. About the ratio between TPU and DP, the relative molecular weight of DP has been given in part 2.2, and the weight of TPU and DP in the hybrids have been given in part 2.3. which wish to be helpful for the readers.
Point 2:On page 4, the authors mentioned that the periodic boundary condition is adoptedin the current work, so why the “free volume” appears to distribute on the inner surface of the cell, according to Fig. 1(h)?
Response 2:Thank you for your suggestion. The corresponding revision has been done in revised manuscript.For MD simulation, althoughthe periodic boundary condition is adopted, visually, only one cell is presented. When the free volume analysis is adopted, the results can only be displayed in the cell. From Fig. 1 (h), The gray ones refer to the free volume only in the cell. The blue ones refer to the free volume in multi cells, which may appear to distribute on the inner surface of the cell.
Point 3:On page 5, for the FTIR figure, Why the authors consider the peak appears at 3143.61, instead of 3000? Besides, I think the references are necessary for the identification of those peaks.
Response 3:Thank you for your suggestion. The corresponding revision has been done in revised manuscript.The FTIR peaks for carboxyl groupappears at about 3200-2500, the peaks for methyl and methylene groups appear at about 3000-2800. For the FTIR figure of AO, to distinguish the carboxyl group and methyl or methylene group, we considered the peak for carboxyl group appear at 3143. 61. However, from 3200-3000 cm-1, two peaks at about 3143 and 3007 are appeared. Therefore, we correct the peak at 3143.61 to the peaks at about 3200-3000, which may be more preciseness. Thank you again for pointing out our mistake. Besides, the references have been added for the identification of those peaks.
Point 4:In Fig. 6(a), I recommend the authors to add the error bar and/or present another work at different DP loading, to get the conclusion of “the FFV showed a drastic decreasing degree from TDP25 to TDP35 compared with that for the low content systems”. In addition, the results of Fig 6(b) and (c) are quite interesting and obviously hard to understand for most readers, so I recommend the authors to explain the underlying reasons, instead of reporting them.
Response 4:Thank you for your suggestion. The corresponding revision has been done in revised manuscript.The error bar has been added in Fig. 6 (a). And the explains for Fig 6 (b) and (c) have been revised.

Round 2
Reviewer 1 Report
The revised manuscript has satisfactorily answered the review comments. It is recommended that the manuscript be accepted for publication in the Journal.
Reviewer 2 Report
The authors have revised thoroughly as my previous comments. Clearly, the quality of the revised manuscript has been improved. Therefore, I suggested it for publication in this journal. There are still some slight grammar errors. The authors should check carefully in the proof.